The leukocyte telomere length, single nucleotide polymorphisms near TERC gene and risk of COPD

Tacheva Tanya tachevat@abv.bg 1
Zienolddiny Shanbeh 2
Dimov Dimo 1
Vlaykova Denitsa 1
Vlaykova Tatyana 1 3
1 Department of Medical Chemistry and Biochemistry, Medical Faculty, Trakia University , Stara Zagora , Bulgaria
2 Section for Toxicology and Biological Work Environment, National Institute of Occupational Health , Oslo , Norway
3 Department of Medical Biochemistry, Medical University - Plovdiv , Plovdiv , Bulgaria
Sjakste Nikolajs
Electronic publication date: 2021 Nov 11
Publication date: 2021
Volume: 9
Electronic Location ID: e12190
Received 2021 Feb 19; Accepted 2021 Aug 30
Copyright: ©2021 Tacheva et al.
Copyright year: 2021
Copyright holder: Tacheva et al.
License: This is an open access article distributed under the terms of the Creative Commons Attribution License, which permits unrestricted use, distribution, reproduction and adaptation in any medium and for any purpose provided that it is properly attributed. For attribution, the original author(s), title, publication source (PeerJ) and either DOI or URL of the article must be cited.
License URL: https://creativecommons.org/licenses/by/4.0/

Keywords: COPD, Telomeres, Telomerase, Polymorphism

Funding: The Medical Faculty, Trakia University, (Stara Zagora, Bulgaria) The present study was supported by project 3/2019 of the Medical Faculty, Trakia University, (Stara Zagora, Bulgaria). The funders had no role in study design, data collection and analysis, decision to publish, or preparation of the manuscript.

==============================
Chronic obstructive pulmonary disease (COPD) is characterized by irreversible airflow obstruction and is associated with chronic local and systemic inflammation and oxidative stress. The enhanced oxidative stress and inflammation have been reported to affect telomere length (TL). Furthermore, a number of SNPs at loci encoding the main components of the telomerase genes, TERT and TERC have been shown to correlate with TL. We aimed to explore the leukocyte TL and genotypes for single nucleotide polymorphisms, rs12696304 (C > G) and rs10936599 (C > T) near TERC in COPD cases and matched healthy controls using q-PCR technologies. Successful assessment of TL was performed for 91 patients and 88 controls. The patients had shorter TL (17919.36 ± 1203.01 bp) compared to controls (21 271.48 ± 1891.36 bp) although not significant (p = 0.137). The TL did not associate with the gender, age, spirometric indexes, smoking habits but tended to correlate negatively with BMI (Rho = − 0.215, p = 0.076) in the controls, but not in COPD patients. The genotype frequencies of the SNPs rs12696304 and rs10936599 were compared between patients and controls and the odds ratios (OR) for developing COPD were calculated. The carriers of the common homozygous (CC) genotypes of the SNPs had higher risk for COPD, compared to carriers of the variants alleles (rs12696304 CG+GG vs. CC; OR: 0.615, 95% CI [0.424–0.894], p = 0.011 and for rs10936599 CT+TT vs. CC OR = 0.668, 95% CI [0.457–0.976], p = 0.044). Analysis on the combined effects of the TERC rs12696304 (C > G) and rs10936599 (C > T) genotypes, CC/CC genotype combination was associated with higher risk for COPD (p < 0.0001) and marginally lower FEV1% pr. in patients with GOLD II (p = 0.052). There was no association between the SNP genotypes and TL. In summary, our results suggest that COPD patients may have shorter TL, and rs12696304 and rs10936599 near TERC may affect the risk of COPD independently of TL.

Introduction

COPD is chronic inflammatory disease characterized by increased mucus production, shortness of breath and cough with sputum production. One of the big challenges in diagnosis and management of COPD comes from the fact that many patients may not recognize their symptoms as being due to the disease but rather due to aging, smoking etc. The pathological changes occur in the proximal and distal airways, as well as in the parenchyma of the lungs. Chronic inflammation, oxidative stress, aging of the cells and the remodeling of the airways are thought to play central role in the pathogenesis of the disease (Chung & Adcock, 2008; GOLD, 2017; Lange et al., 2016; Pauwels & Rabe, 2004).

Cellular aging, or senescence, results in series of alterations in cell morphology and function, including the loss of proliferative activity (MacNee, 2009). Aging of somatic cells results in a depletion of their potential of division due to shortening of telomeres or in response to extra- or intracellular stress factors (Savale et al., 2009).

There is much evidence of common features between pulmonary emphysema and lung aging as oxidative stress, inflammation and apoptosis are common for both (MacNee, 2009). The number of aging lung vascular endothelial cells is higher in COPD patients as it has been thought decreased telomerase activity and shortened telomeres are the main factor in this process (Amsellem et al., 2011).

Telomeres are specialized nucleoprotein structures at chromosome ends. In vertebrates they consist of tandem repeats of TTAGGG sequence stabilized by a complex of proteins, known as shelterin, protecting the natural chromosome termini from the DNA damage signaling and repair activities (Bertuch, 2016; Birch et al., 2015). The critical minimum number of 12.8 telomere repeats is required to protect chromosome from recombination and fusion (Bernadotte, Mikhelson & Spivak, 2016). DNA replication is incomplete at the 3′ end of the linear chromosome, due to the end replication problem (Maubaret et al., 2013). Thus telomere length (TL) decreases 50 to 200 base pairs with each cell division until after 50 to 60 cell cycles they reach the critical length (the Hayflick limit). As concequence, the cell cycle is interrupted resulting in the cell-cycle arest and the cell undergoes apoptosis or becomes senescent (Aix et al., 2016; Alter et al., 2007; Bertuch, 2016; Fumagalli et al., 2014; Maubaret et al., 2013). Telomeres can also be shortened as a result of DNA double-strand break damage (Bernadotte, Mikhelson & Spivak, 2016).

Telomere shortening is associated with organism aging, as leukocytes telomere length (LTL) is inversely related to age and is associated with increased risk of age-related diseases and with mortality (Soerensen et al., 2012). Telomerase is a specialized reverse transcriptase composed of catalytic protein (reverse transcriptase, TERT) and telomerase RNA component (TERC). The latter is the constituent of the enzyme, which provides the template for the synthesis of the TTAGGG repeats. The enzyme telomerase prevents the shortening of the telomeres by synthesizing telomeric repeats on to the end of the 3′  G-rich strand. To prolong their growth capacity, highly dividing cells, such as stem cells, maintain high activity of this enzyme complex. A number of short nucleotide polymorphisms (SNPs) at loci encoding the main components of the telomerase genes, TERT and TERC have been shown to correlate with the TL (Bertuch, 2016; Calado, 2009; Maubaret et al., 2013; Schmidt & Cech, 2015).

It has been found that in COPD patients there is a low activity of the telomerase (Kordinas, Ioannidis & Chatzipanagiotou, 2016) and shorter telomeres of the chromosomes of the circulating leukocytes, alveolar epithelial cells and lung vascular endothelial cells (Amsellem et al., 2011; Houben et al., 2009; Savale et al., 2009), as well as of the senile smooth muscle cells in remodeled blood vessels (Morla et al., 2006).

In this context, the identification of SNPs in the pathogenesis of COPD and the establishment of potential biomarkers like LTL for predicting clinical outcomes in COPD phenotypes would be helpful for personalizing the management of the disease.

Materials and Methods

Patients and controls

All together 194 patients with COPD and 281 non-affected control individuals from the region of Stara Zagora, Bulgaria, were included in our case-control study. The including criteria for the individuals with COPD was as described previously (Tacheva et al., 2017). The patients were recruited in the Clinic of Internal Medicine, University Hospital, Trakia University, Stara Zagora, Bulgaria from 2008 till 2011. Patients had different stages of the disease according to GOLD (GOLD II, III and IV). The inclusion criteria for COPD were: age higher than 40 years; forced expiratory volume in 1 s (FEV1) of <80%; forced expiratory volume in 1 s (FEV1)/forced vital capacity (FVC) ratio of ≤ 70%; FEV1 reversibility after inhalation of Salbutamol 400 µg of <12%.

The control group consisted of 281 healthy unaffected by any lung diseases volunteers from the same ethnic group and area of Bulgaria.

The available demographic and clinical data of both COPD patients and control individuals are presented in Table 1.

Table 1 Demographic and clinical data of COPD patients and controls.

	COPD patients	Controls	
	rs12696304
(C > G)	rs10936599
(C > T)	LTL	rs12696304
(C > G)	rs10936599
(C > T)	LTL	
Number
males
females	(n = 189)
142 (75.1%)
47 (24.9%)	(n = 191)
143 (74.9%)
48 (25.1%)	(n = 91)
67 (73.6%)
24(26.4%)	(n = 277)
128(46.2%)
149(53.8%)	(n = 263)
121 (46%)
142 (54%)	(n = 88)
38 (43.2%)
50 (56.8%)	
Age at the inclusion in the study							
median(range) (years)	67 (36–88)	67 (36–88)	66 (40–88)	57 (19–86)	56 (19–86)	57 (23–79)	
Age at the diagnosis of the disease							
median (range) (years)	64 (30–86)	62 (30–86)	63 (37–86)				
Duration of the disease							
mean ± SD (years)	4.9 ± 5.4	4.9 ± 5.4	4.7 ± 4				
median (range) (years)	3 (0–30)	3 (0–30)	3.5 (0–20)				
Smoking status
non-smokers
ex-smokers
current smokers	(n = 185)
54 (29.2%)
87 (47.0%)
44 (23.8%)	(n = 192)
55 (28.6%)
89 (46.4%)
48 (25%)	(n = 89)
29 (32.6%)
41 (46.1%)
19 (21.3%)	(n = 191)
113 (59.2%)
30 (15.7%)
48 (25.1%)	(n = 186)
110(59.1%)
28 (15.1%)
48 (25.8%)	(n = 73)
47(64.4%)
7 (9.6%)
19 (26%)	
Smoking habits (packs/year)							
mean ± SD (years)	31.8 ± 15.2	31.8 ± 15.2	29.2 ± 14	19.1 ± 12.1	18.9 ± 12.2	22.8 ± 14.2	
median (range)	30 (5–88)	30 (5–88)	30 (5–60)	20 (5–60)	20 (5–60)	20 (5–60)	
FEV1% pr.							
mean ± SD	51.7 ± 13.6	51.7 ± 13.6	51 ± 13.7	97 ± 14.7	97 ± 14.7	95.2 ± 13	
FEV1/FVC %							
mean ± SD	61.3 ± 8.5	61.3 ± 8.5	62 ± 8.1	80 ± 8.7	80 ± 8.7	80.6 ± 8.3	
COPD stage (GOLD 2009)
GOLD II
GOLD III
GOLD IV	(n = 189)
106 (56.1%)
72 (38.1%)
11 (5.8%)	(n = 191)
107 (56%)
73 (38.2%)
11 (5.8%)	(n = 91)
52 (57.1%)
32 (35.2%)
7 (7.7%)				

The Ethics committee at Medical Faculty, Trakia University, Stara Zagora, Bulgaria has approved the study protocol and written informed consents were obtained from all participants before the study.

Isolation of DNA

For the isolation of genomic DNA a commercial kit (GenElute™ Mammalian Genomic DNA Miniprep Kit, Sigma, USA) was used. The purity of the isolated DNA and the concentration (in ng/µl) were assessed spectrophotometrically (NanoDrop Spectrophotometer ND-1000, Thermo Fisher Scientific - NanoDrop products, NanoDrop Technologies Inc). The purity was evaluated by calculating the ratio of the OD at λ260 nm and λ280 nm.

Genotyping

The genotyping for TERC C > G and TERC C > T SNP was performed by TaqMan q-PCR. The PCR mix contained 1.5 µl TaqMan Genotyping Master Mix (Applied Biosystems, USA), 0.1 µl 40x SNP Genotyping assay with VIC® and FAM™ dyes, 1 µl genomic DNA (10 ng/µl) and distilled water up to the final volume of 4 µl. The reaction was performed with preliminary denaturation and activation of the polymerase for 2 min at 50 °C and 10 min at 95 °C, followed by 40 cycles of denaturation at 95 °C for 15 s (slope, 20 °C/s), annealing and primer extension at 60 °C for 1 min. The data from the fluorescence of the both dyes were detected by 7900 Fast Real-Time PCR System (Applied Biosystems, USA). The Allele discrimination software was used for determining the genotypes of the studied persons.

Leukocyte telomere length measurement

The telomere length was measured using SYBR green qPCR method (O’Callaghan et al., 2008). The method is based on using a standard curve made of serial dilutions of a pTEL plasmid containing a 600 base pair long telomere sequence. The telomeres were amplified by using primers that bind to the telomere region of the template DNA. By using the standard curve, each sample is given a quantity, which corresponds to the amount of kilo bases of telomere sequence in the reaction. Fth1 was used as a reference gene.

The DNA samples were diluted to a final concentration of 0.5 ng/µl. For each reaction, 2 µl DNA, 5 µl SYBR green, 2.6 µl H2O, and 0.2 µl of each primer were mixed to a total of 10 µl per well, and the samples were run as duplicates. The thermocycler that was used was Applied Biosystems’ 7900 real-time machine. The thermal profile of the qPCR reaction was as follows: preliminary denaturation and activation of the polymerase at 95 °C for 2 min, followed by 40 cycles of denaturation at 95 °C for 10 s and annealing and primer extension at 60 °C for 45 s, with a dissociation-curve: 95 °C for 15 s, 60 °C for 15 s, 95 °C for 15 s.

Statistical analyses

Statistical analyses were performed using SPSS 16.0 for Windows (IBM, Chicago,IL). The continuous variables with normal distribution were compared between two or more independent groups by Student t-test or One-way ANOVA test with LSD Post hoc analysis, while those variables with non-normal distribution were compared by using Mann–Whitney U test or Kruskal–Wallis H test, respectively. The correlations between the continuous variables were assessed using the Pearson or Spearman correlation tests according to the type of the variables’ distribution.

By using the χ2 test, the genotype distributions of control individuals and COPD patients were tested for deviation from Hardy-Weinberg equilibrium (HWE). The differences in genotype and allele distributions between the groups were analyzed in 2×2 or 2×3 contingency tables with χ2 test. Binary Logistic regression was applied for calculation of the odds ratios (OR) and 95% confidence interval (CI) with age and sex as covariates. Factors with p < 0.05 were considered statistically significant.

Results

TERCrs12696304 (C > G) SNP

The genotype frequencies according to rs12696304 (C > G) polymorphism in both groups did not deviate from Hardy–Weinberg equilibrium (p = 0.953 for COPD patients and p = 0.531 for the control individuals).

In the genotype distribution we found statistical difference between controls and COPD patients (p = 0.035, χ2 test). The genotype frequencies of both groups are presented in Table 2. In the allele distribution there was also a significant difference (p = 0.019, Table 2). The results of the logistic regression analysis showed that the carriers of CG genotype have 1.58-fold lower risk of developing COPD compared to the individuals with homozygous CC genotypes (p = 0.020, Table 2). The significance remained also after adjustment for sex and age (p = 0.028, Table 2). In the recessive model, the carriers of genotypes containing at least one variant G allele (CG+GG) had 1.63-fold lower risk of COPD than those with CC genotypes (p = 0.011, Table 2). The significance remained also after adjustment for sex and age (p = 0.022, Table 2).

Table 2 Genotype and allele distributions according to TERC C > G (rs12696304) SNP in COPD patients and controls.

rs12696304 (C > G)	COPD patients	Controls	OR (95% CI), p-value	OR (95% CI), p-value adjusted for sex and age ≥60 years)	
	n	frequency	n	frequency			
	n = 189		n = 277				
Genotype frequency	
CC	97	0.513	109	0.393	1.0 (referent)		
CG	78	0.413	139	0.502	0.631 (0.427–0.931), p = 0.020	0.621 (0.407–0.949), p = 0.028	
GG	14	0.074	29	0.105	0.542 (0.271–1.086), p = 0.084	0.652(0.294–1.328), p = 0.222	
CG+GG	92	0.487	168	607	0.615 (0.424–0.894), p = 0.011	0.622 (0.414–0.933), p = 0.022	
Allele frequency	
rs12696304 C	272	0.720	357	0.664	1.0 (referent)		
rs12696304 G	106	0.280	197	0.356	0.706 (0.532–0.938), p = 0.019		

TERC C > T (rs10936599) SNP

The genotype distribution according to TERC C>T (rs10936599) SNP was in equilibrium with the Hardy–Weinberg principle (p = 0.272 for controls and p = 0.808 for COPD patients). The frequency of genotypes in control individuals and COPD patient can be seen in Table 3. The genotype distribution in COPD patients did not differ from that in controls (p = 0.113, Table 3). A marginal difference was found in the allele distribution—the variant T allele was more common in controls than in patients (p = 0.058, Table 3)

Table 3 Genotype and allele distributions according to TERC C > T (rs10936599) SNP in COPD patients and controls.

rs10936599 (C > T)	COPD patients	Controls	OR (95% CI), p-value	OR (95% CI), p-value adjusted for sex and age ≥ 60 years)	
	n	frequency	n	frequency			
	n = 191		n = 263				
Genotype frequency	
CC	119	0.623	138	0.525	1.0 (referent)	1.0 (referent)	
CT	66	0.346	114	0.433	0.671 (0.455–0.992), p = 0.045	0.683 (0.447–1.042), p = 0.077	
TT	6	0.031	11	0.042	0.633 (0.227–1.762), p = 0.381	0.587(0.197–1.751), p = 0.339	
CT+TT	72	0.377	125	0.475	0.668 (0.457–0.976), p = 0.044	0.674 (0.446–1.016), p = 0.060	
Allele frequency	
rs10936599 C	304	0.796	390	0.741	1.0 (referent)		
rs10936599 T	78	0.204	136	0.259	(0.537–1.009), p = 0.058		

The carriers of the more common homozygous CC genotype had 1.5-fold higher risk for COPD compared to the heterozygous (p = 0.045) and than those carrying the variant T allele (CT+TT) (p = 0.044, Table 3).

Analysis on the combined effects of the TERC rs12696304 (C > G) and rs10936599 (C > T) genotypes on the development of COPD showed that, individuals who are carriers simultaneously of CC/CC genotypes have 1.98-fold higher risk for COPD compared to persons having all other genotype combinations (p < 0.0001) (Table 4). This very strong significance remained also after adjustment for gender and age (p = 0.001) (Table 4). The spirometric indices of the patients carrying CC/CC genotypes did not differ significantly than those with all other genotype combinations (FEV1% pr. 50.35 ± 1.34 (SEM) vs. 53.05 ± 1.52 (SEM), p = 0.185; FEV1/FVC 61.61 ± 0.86 (SEM) vs. 61.00 ± 0.93 (SEM), p = 0.626). However, when the patients were dichotomized into two groups (GOLD stage II and GOLD stage III/IV) we found that CC/CC genotype combination was associated with marginally lower FEV1% pr. than the other genotype combinations only in patients with moderate COPD (GOLD II) (58.87 ± 1.10 (SEM) vs. 62.45  ± 1.46, p = 0.052). Such difference was not observed between the genotype carriers in patients with severe/very severe COPD (GOLD stage III/IV), (39.00 ± 1.35 (SD) vs. 40.75  ± 1.54, p = 0.364).

Table 4 Genotype combinations according to the two studied SNPs rs12696304 (C > G) and rs10936599 (C > T) and the frequency in controls and COPD patients.

rs12696304 (C > G) and rs10936599 (C > T)	COPD patients	Controls	OR (95% CI), p-value	OR (95% CI), p-value adjusted for sex and age ≥60 years)	
	n	frequency	n	frequency			
	n = 190		n = 278				
Genotype combination frequency	
CC/CC	97	0.511	96	0.345	1.0 (referent)	1.0 (referent)	
All other genotype combinations	93	0.489	182	0.655	0.506 (0.347–0.737), p < 0.0001	0.507 (0.337–0.764), p = 0.001	

Leukocyte telomere length

Comparison of the leukocyte telomeres showed that they are shorter in COPD patients compared to controls, but without significant difference (17919.36 ± 1203.01 (SEM) bp vs. 21271.48 ± 1891.36 (SEM) bp, p = 0.137) (Fig. 1). When splitting controls and COPD patients into two subgroups - less than and above 60 years of age, we found near-marginal significance for shorter telomeres in COPD patients above 60 years compared to controls in the same subgroup (17558.73 ± 1536.78 (SEM) bp vs. 22257.88 ± 3684.15 (SEM) bp., p = 0.188, Fig. 2), but no difference was seen in younger subgroups (p = 0.479).

Figure 1 Leukocyte telomere length (LTL) in COPD patients and controls.

The LTL is presented as mean ± standard error of mean (SEM).

Figure 2 Leukocyte telomere length (LTL) in COPD patients and controls less than and above 60 years of age.

The LTL is presented as mean ± standard error of mean (SEM).

In COPD patients we found no differences in the LTL between GOLD stages (p = 0.468, ANOVA test). There were also no statistically significant correlations between the LTL and age p = 0.903), age of disease diagnosis (p = 0.962) and spirometric indices of the lung function FEV1% (p = 0.814) and FEV1/FVC% (p = 0.631). LTL did not differ between the male and female COPD patients (p = 0.169), as well as when compared between the patients with different BMI (normal BMI vs. overweight vs. obese, p = 0.398). Similarly, in control group, LTL did not differ between sexes (p = 0.835), and did not correlate with the age (p = 0.759) and spirometric indices (p = 0.991 for FEV1% pr, and p = 0.419 for FEV1/FVC). There was only a tendency for a weak negative correlation between LTL and BMI in controls (Rho = −0.215, p = 0.076). When the LTL were compared between the controls with normal BMI vs. overweight vs. obese, in the LSD post hoc analysis of ANOVA test the obese controls had marginally lower LTL (14752.22 ± 2938.48 (SEM) bp) than the controls with normal BMI (23008.93 ± 1867.79 (SEM) bp, p = 0.091).

Unexpectedly, the smoking habits did not influence the LTL either in patients (current smokers: 17238.96 ± 2853.24 (SEM) bp vs. ex-smokers: 19214.10 ± 1790.28 (SEM) bp vs. non-smokers: 16700.79 ± 2127.42 (SEM) bp, p = 0.644, ANOVA test) or in controls (current smokers: 19393.32 ± 2965.92 (SEM) bp. vs. ex-smokers: 17999.97 ±3160.78 (SEM) bp. vs. non-smokers: 22676.26 ± 3086.59 (SEM) bp., p = 0.712, ANOVA test). In COPD patients aged above 60 years which also are current/ex-smokers, telomeres were shorter in man compared to women (14909  ± 1957.87 (SEM) bp vs. 30925 ± 6818.35 (SEM) bp, p = 0.006).

We found no association between LTL and the studied polymorphisms rs12696304 and rs10936599 near TERC, both in controls and COPD patients (Table 5).

Table 5 Leukocyte telomere length (LTL) in COPD patients and controls.

	COPD patients	Controls	
	n	LTL mean (bp)	LTL SEM (bp)	p-value	n	LTL mean (bp)	LTL SEM (bp)	p-value	
rs12696304 (C > G)	88			0.847	84			0.459	
CC	54	17626	1543.44	referent	47	23749	3176.03	referent	
CG	30	18217	1024.81	0.818*	28	18540	1885.04	0.226*	
GG	4	15164	5083.14	0.673*	9	19964	4445.9	0.562*	
rs10936599 (C > T)	90				86			0.925	
CC	64	17809	1435.59	referent	49	21787	3016.24	referent	
CT	24	18413	2464.21	0.828*	31	20199	2204.13	0.703*	
TT	2	11011	1935.76	0.416*	6	20359	6633.99	0.859*	
rs12696304 (C > G) and rs10936599 (C > T)	88				84				
CC/CC	34	17926	1543.44	referent	42	22988	3438.71	referent	
All other genotype combinations	54	17858	1866.73	0.925	42	20226	1850.85	0.482	
Notes.

* LSD post hoc analysis of ANOVA test.

Discussion

In our study, we found shorter LTL in patients with COPD, even though without significance. The absence of statistical difference might be due to the small number of both groups (88 control individuals and 91 patients with COPD). Nevertheless our results are with accordance to others where it has been found an association of telomere shortening with COPD in circulating leukocytes (Savale et al., 2009), alveolar epithelial cells (Houben et al., 2009; Mui et al., 2009), and pulmonary vascular endothelial cells (Amsellem et al., 2011). In addition shorter telomeres in the lungs of individuals with emphysema and pulmonary fibrosis have been found when compared with unaffected individuals (Alder et al., 2015; Alder et al., 2011; Tsuji, Aoshiba & Nagai, 2006).

Human leukocytes telomeres have been shown to decrease approximately with 20–30 bp per year in vivo (Daniali et al., 2013). Besides genetic factors there are variety of environmental and behavior ones (tobacco smoking, diet, stress, air pollution etc.) which have influence on leukocytes telomere (Liu et al., 2019). Many of these factors have been shown to be associated with higher risk of COPD.

It seems in our groups those factors do not independently have an impact on LTL but when combining they may reflect on the length of the telomeres. Although our subgroup in this case is quite small we found significantly shorter telomeres in male current/ex-smokers COPD patients above 60 years than female from the same subgroup. Besidest smoking it has been show that sex might also be an important factor in telomere shortening. Longer telomeres in women might be due to estrogen which by binding to estrogen response element in the hTERT promoter region contribute to telomere restoration (Mayer et al., 2006). It has been found that oxidative stress also plays role in telomere shortening. By reducing the production of ROS and being a potent antioxidant and regulator of antioxidant genes, estrogen might affect the TL (Lulkiewicz et al., 2020).

It is known that in COPD oxidative stress is of great importance for the disease development and progression. Several animal and in vitro models have shown that chronic oxidative stress induces an accelerated rate of telomere loss. Other pathological conditions such as atherosclerosis, diabetes type I (Houben et al., 2009) cardiovascular disease (Fitzpatrick et al., 2011), renal failure (Boxall et al., 2006), various cancers (Broberg et al., 2005), Alzheimer’s disease (Cai, Yan & Ratka, 2013), and Parkinson’s disease (Maeda et al., 2012) have been related to constant high level of oxidative stress and chronic inflammation and linked with telomere shortening (Kordinas, Ioannidis & Chatzipanagiotou, 2016). Additionally, telomere length can be dynamically changed throughout an individual’s life period in response to environmental factors and stress (Aviv, 2002; Carlson et al., 2015).

The process of senescence in somatic cells occurs in result of depletion of the replicative potential or in response to excessive extracellular or intracellular stress (Campisi & d’Adda di Fagagna, 2007), as both forms of senescence may be present in COPD (Savale et al., 2009; Tsuji, Aoshiba & Nagai, 2004).

Amsellem et al. have found increased number senescent smooth muscle cells in the media of remodeled vessels from patients with COPD. Premature senescence in COPD has been associated with overexpression of pro-inflammatory cytokines and adhesion molecules, together with shortened telomeres (Amsellem et al., 2011).

In their study Alder et al. (2015) highlight the contrast in the response to telomere dysfunction between high- and low turnover tissues. Similarly, human telomere-mediated lung diseases become symptomatic in late adulthood, reflecting a slower development of the process (Parry et al., 2011).

The senescence is associated with accumulation of cyclin-dependent kinase inhibitors (Tsuji, Aoshiba & Nagai, 2006). Telomeric signals are mediated mainly by the p53–p21 pathway and non-telomeric signals by the p16–retinoblastoma protein pathway (Campisi & d’Adda di Fagagna, 2007). In the lungs of patients with emphysema increased numbers of p21- and p16-stained cells have been found (Amsellem et al., 2011). Dysfunctional telomeres activate a DNA damage response, trigger the formation of anaphase bridges, and up-regulate the cell-cycle inhibitor p21 (Aix et al., 2016).

Alterations of airway epithelium, including squamous cell metaplasia, goblet and basal cell hyperplasia, are often present in smokers with COPD. Differentiation of variety of tissue types, including squamous epithelia, is associated with an increased p21Waf1 expression. The cytoplasmic and nuclear p-21 immunoreactivity has been found significantly increased in COPD smokers compared to controls (Chiappara et al., 2013). In contrast to our results, Birch et al. (2015) failed to detect differences in TL between controls and patients with COPD. Consistently, another study also failed to find differences in TL between lung fibroblasts isolated from patients with emphysema and aged-matched controls, despite increased expression of senescence-associated markers (Muller et al., 2006). Interestingly, it has been found that the median telomere length is longer in peripheral blood leukocytes from COPD patients with a1-antitrypsin deficiency compared with COPD controls (Saferali et al., 2014).

The differences in the results reported in variety of studies on TL in chronic lung diseases might be due to the different methods used for telomere length measurement. On the other hand the length of telomeres varies between individuals and the different cells and chromosomes in the same person. Another fact that should not be underestimated is that LTL might be altered by therapeutic interventions (Lulkiewicz et al., 2020).

After all the factors (endogenous mechanisms and exogenous factors) resulting in TL changes are quite diverse and many studies should be done in order to understand the exact mechanism of TL regulation.

Some common SNPs near TERC have been found to be involved in telomere biology including rs12696304 (C > G) and rs10936599 (C > T) (Codd et al., 2010; Codd et al., 2013; Levy et al., 2010; Shen et al., 2011; Vasa-Nicotera et al., 2005). In a genome-wide association study of mean leukocyte telomere length in 2,917 individuals, it has been found that each rs12696304 (G) allele is significantly associated with reduction in the mean telomere length (Codd et al., 2010). The minor alleles for both SNPs have been found to negatively correlate with LTL (Michalek et al., 2017; Soerensen et al., 2012).

Interestingly, opposite to our findings of shorter LTL in patients with COPD, we obtained higher frequency of the referent allele of both SNPs (rs12696304 and rs10936599) in the patients group. It appeared that those genotypes containing the variant allele (rs12696304 CG+GG and rs10936599 CT+TT) are protective. The results on the role of rs12696304 and rs10936599 SNPs in chronic inflammatory diseases, which we can compare ours with, are quite diverse and ambiguous. For example, in unison with our findings of higher frequency of the referent allele in the group of patients, a significant association of the major allele (C) of rs10936599 with the high risk for type 2 diabetes has been reported (Sethi et al., 2020). On the other hand in a study of the associations of TERC SNPs with LTL and the risk of type 2 diabetes mellitus in Kuwaiti population it has been found that homozygous carries of less common allele (G) of rs12696304 have shorter LTL compared with other genotypes. Besides having the shortest LTL homozygous carriers of the minor allele G appeared to have significantly lower hTERT serum levels and highest insulin resistance compared to the other two genotypes (Al Khaldi et al., 2015). Similar findings for shorter telomeres in carriers of the minor G allele of rs12696304 have been reported in European, American and Chinese Han populations (Codd et al., 2010)

Higher frequency of rs12696304 G allele have been found in individuals with chronic kidney disease with no difference of LTL among different genotypes (Sun et al., 2020).

In another chronic inflammatory lung disease - Coal workers’ pneumoconiosis and in Coronary heart disease (CHD) no association with rs12696304 and rs10936599 SNPs has been found (Maubaret et al., 2013; Yuan et al., 2020).

It is clear that in most of the studies the minor alleles are related to shorter telomeres and pathogenesis and clinical outcomes of chronic inflammatory diseases. Still, the presence of reports showing no association of the genotype with the TL in chronic diseases suggest that other mechanisms might play role in the regulation of telomere length.

In telomerase-null mice, DNA damage appears in the air-exposed epithelium after environmentally induced injury (cigarette smoke for example). The additive effect of environmental injury and telomere dysfunction has been suggested to contribute to the susceptibility to emphysema seen in these mice (Alder et al., 2015).

In addition histone methylation in the telomere region and demethylation of the human TERT both play significant role in maintaining heterochromatin, transcription silencing at telomeres, and telomerase inactivation (Bernadotte, Mikhelson & Spivak, 2016).

According to our study, the mean LTL is shorter in patients with COPD, but it is not related to the genotype. Perhaps, other factors which play role in the development of the disease (chronic inflammation and oxidative stress), may influence the telomere length. Because the main limitation of the present study is the small sample size of both COPD and controls groups, more studies with significantly bigger in size groups have to be done in order to reveal the role of telomeres and the polymorphisms in TERC in the development and progression of COPD.

Supplemental Information

Supplemental Information 1 Raw data

Click here for additional data file.

Additional Information and Declarations

Competing Interests

Author Contributions

Human Ethics

Data Availability

The authors declare there are no competing interests.

Tanya Tacheva performed the experiments, analyzed the data, prepared figures and/or tables, authored or reviewed drafts of the paper, and approved the final draft.

Shanbeh Zienolddiny conceived and designed the experiments, analyzed the data, authored or reviewed drafts of the paper, and approved the final draft.

Dimo Dimov conceived and designed the experiments, analyzed the data, prepared figures and/or tables, and approved the final draft.

Denitsa Vlaykova analyzed the data, prepared figures and/or tables, and approved the final draft.

Tatyana Vlaykova conceived and designed the experiments, performed the experiments, analyzed the data, prepared figures and/or tables, authored or reviewed drafts of the paper, and approved the final draft.

The following information was supplied relating to ethical approvals (i.e., approving body and any reference numbers):

The study protocol was approved by the ethics committee at Medical Faculty, Trakia University, Stara Zagora, Bulgaria and informed consents were obtained from all participants before the study.

The following information was supplied regarding data availability:

The raw data is available in the Supplemental Files.

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
