# Peer review of "The leukocyte telomere length, single nucleotide polymorphisms near TERC gene and risk of COPD"

_PeerJ, doi:10.7717/peerj.12190_

## Round 0.1 · original submission · Major Revisions

Dear authors,
Your manuscript was reviewed quickly but thoroughly. Please edit your manuscript following the comments of the Reviewers.
Nikolajs

Reviewer 1 ·

Basic reporting

Article analyses

Authors aimed to explore the leukocyte TL and genotypes for single nucleotide polymorphisms rs12696304 (C>G) and rs10936599 (C>T) near TERC in COPD cases compared with healthy individuals. The topic is interesting and relevant. Manuscript is well structured.
English grammar and style should be improved in some places (e.g. line 178 the smoking habits did not influenced the LTL”).
In figure 1 it is written "the LTL is presented as mean ± standard deviation (SEM)", but only mean is presented.
Please format table 1.
The information in lines 124-129 and 140-145 repeats the information which is provided in table 2 and table 3. I suggest mentioning only significant results and do not rewrite figures in the text, because they can be seen in the tables.
Lines 166-167 provide the same information as the figure.

Experimental design

The topic is relevant. The materials and methods part is described clearly.

Validity of the findings

In lines 178-179 authors write “the smoking habits did not influence the LTL either in patients (p=0.644, 179 ANOVA test) or in controls (p=0.712, ANOVA test)”. Could you clarify between which groups you compared LTL length (current smokers vs. nonsmokers separately in COPD patients and healthy subjects?) and could you provide the LTL length in these groups.
In the discussion part authors focus mainly on LTL length but I missed more discussion about single nucleotide polymorphisms near TERC gene. Could you extend the discussion in this direction. I suggest to discuss and compare your results with the results from research which analyze SNP near TERC in other diseases which main pathogenesis is related with chronic inflammation and oxidative stress.

Reviewer 2 ·

Basic reporting

Comments to the Authors
Summary: The manuscript is about telomere length and SNPs in Chronic obstructive pulmonary disease (COPD) patients. Authors have found no statistical difference between LTL in COPD patients` and controls¬` group, although for the patients` telomeres were slightly shorter. They found that the carriers of the common homozygous genotype of the SNPs (rs12696304 C and rs10936599 C) had higher risk for COPD in comparison with other allele variants, which is an interesting observation that the minor alleles have rather protective properties, and it could be studied further.

Recommendation for the corrections:

1. Introduction.
a) Line 34 needs correction of punctuation mistakes. Needs precision of what is the minimum number of telomere repeats.
b) Missing a clear hypothesis of the study.

Experimental design

2. Methods.
a) The description of samples also could be clearer. As it is not clear why for example only a small part of samples was used for the LTL determination. Why is there a difference for an average age of about 10 years for the control group and patients? As for TL there is a difference in these two age groups in the literature.
b) Table 1 and other Tables too must be more in order so that text is aligned everywhere and are not so chaotic. Table 4 has a mistake, is it 19 or 190 samples?
c) Regarding qPCR for TL determination. Authors have measured absolute TL, but that number is very high (healthy people in average 21 271 bp). There is information in the literature showing that an average telomere length ranges from 10 to 12 Kbp in new-borns to 5 to 6 Kbp in people older than 60 years. The method for the TL determination should be reconsidered and rather relative TL should be measured (Cawthon, 2002).
d) Also, in description of qPCR for TL and reference genes Efficiency should be determined (and should be from 95-105%). Samples are better to run in triplicates not only in duplicates as it is a rather sensitive method.

Validity of the findings

3. Results.
a) The first two titles for the subsections should be written in synchronized form.
b) It could be interesting to see if there is a difference in TL among those three COPD stages, although it is clear that the sample amount is small for the last stage.
c) As I wrote before the assay for measuring should be considered as authors did not see difference in age, sex and smoking habits which is rather unusual as in the literature these are factors that clearly affects TL.

4. Discussion.
a) Text is a bit chaotic which makes some parts of the text seem redundant.
b) Lines 221 and 222 need punctuation correction.
c) How can authors conclude that LTL is shorter in patients with COPD, but it is not related to the genotype? I did not find any description in the results about TL in different genotypes. I don`t think that conclusion can be drawn only from a fact that they obtained higher frequency of the referent allele of both SNPs (rs12696304 and rs10936599) in the patients` group with shorter telomeres, especially if the difference was not significant and samples were not that many.

---

## Round 0.2 · accepted · Accept

You have thoroughly edited the manuscript following recommendations of the Reviewers. The paper is publishable now.

Reviewer 1 ·

Basic reporting

Authors corrected manuscript according to the comments and in my opinion it meets the PeerJ criteria

Experimental design

Authors corrected manuscript according to the comments and in my opinion it meets the PeerJ criteria

Validity of the findings

Authors corrected manuscript according to the comments and in my opinion it meets the PeerJ criteria